# Extracting Turnover Frequencies of Electron Transfer in Heterogeneous Catalysis: A Study of $IrO_2$-$TiO_2$ Anatase for Water Oxidation Using $Ce^{4+}$ Cations

Mogbel Alrushaid [1], Muhammad A. Nadeem [1], Khaja A. Wahab [1] and Hicham Idriss [1,2,*]

[1] Surface Science and Advanced Characterizations Department, SABIC-CRD at KAUST, Thuwal 23955, Saudi Arabia; Rushaidma@SABIC.com (M.A.); NadeemMI@SABIC.com (M.A.N.); kwahabahmed@googlemail.com (K.A.W.)

[2] Department of Chemistry, University College London, London WC1H 0AJ, UK

[*] Correspondence: idrissh@sabic.com or h.idriss@ucl.ac.uk

**Abstract:** Within the context of electron transfer during the catalytic water oxidation reaction, the Ir-based system is among the most active. The reaction, mimicking photosynthesis II, requires the use of an electron acceptor such the $Ce^{4+}$ cation. This complex reaction, involving adsorbed water at the interface of the metal cation and $Ce^{4+}$, has mostly been studied in homogenous systems. To address the ambiguity regarding the gradual transformation of a homogenous system into a heterogeneous one, we prepared and studied a heterogeneous catalyst system composed of $IrO_2$, with a mean particle size ranging from about 5 Å to 10 Å, dispersed on a $TiO_2$ anatase support, with the objective of probing into the different parameters of the reaction, as well as the compositional changes and rates. The system was stable for many of the runs that were conducted (five consecutive runs with 0.18 M of $Ce^{4+}$ showed the same reaction rate with TON > 56,000) and, equally importantly, was stable without induction periods. Extraction of the reaction rates from the set of catalysts, with an attempt to normalize them with respect to Ir loading and, therefore, to obtain turnover frequencies (TOF), was conducted. While, within reasonable deviations, the TOF numbers extracted from TPR and XPS Ir4f were close, those extracted from the particle shape (HR-STEM) were considerably larger. The difference indicates that bulk Ir atoms contribute to the electron transfer reaction, which may indicate that the reaction rate is dominated by the reorganization energy between the redox couples involved. Therefore, the normalization of reaction rates with surface atoms may lead to an overestimation of the site activity.

**Keywords:** oxygen evolution reaction; OER; $Ce^{4+}$/$Ce^{3+}$; $IrO_2$/$TiO_2$; water oxidation; scanning transmission electron microscopy (STEM); X-ray photoelectron spectroscopy (XPS Ir4f); metal clusters; iridium dispersion; turnover frequency (TOF)

## 1. Introduction

The abundance of water on earth and the mature, yet still expensive, electrolysis systems have led to extensive research targeting other water splitting routes as a source of renewable hydrogen [1,2]. While using sunlight to generate $H_2$ is a plausible scenario in terms of overcoming global energy problems, the overall catalytic process was found to be overwhelmingly challenging. While splitting this process into two half reactions, an oxygen evolution reaction (OER) and a hydrogen evolution reaction (HER), cannot address the overall water splitting, it may offer some insights, and therefore fundamental knowledge, on electron transfer reactions [3–5]. The complexity of the four-electron transfer reaction (OER) motivated many researchers working on homogenous (molecular) systems to obtain structural, mechanistic and electronic information [5–9].

Cerium ammonium nitrate (CAN; $Ce^{4+}$ cations), as a single electron oxidant, has been widely used as a sacrificial agent to study OER on homogenous Ir, Ru, (and, in some cases, Mn or Fe)-based catalysts.

The redox potential level of the $Ce^{4+}/Ce^{3+}$ pair thermodynamically drives water oxidation reactions. The reaction takes place according to $2H_2O + 4Ce^{4+} \rightarrow 4Ce^{3+} + O_2 + 4H^+$ [5]. In the presence of Ir-based catalysts, the enhancement of the OER reaction rate was linked to the formation of Ir(V) species as an intermediate species. For example, Minguzi and co-workers experimentally demonstrated the formation of Ir(V) during OER activity of Ir using in-situ X-ray absorption spectroscopy [7]. The existence of the $Ir^{5+}/Ir^{4+}$ redox potential between that of $Ce^{4+}$ reduction and water oxidation makes it suitable. Scheme 1 presents a simplified redox potential requirement for the possible catalytic reaction of the $Ir^{4+}/Ce^{4+}/H_2O$ system.

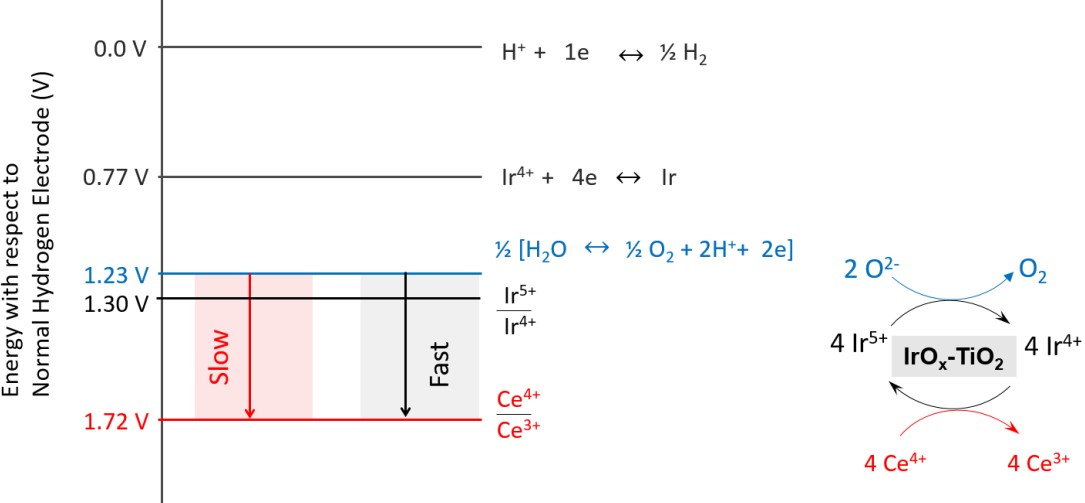

**Scheme 1.** A schematic representation of the catalytic and non-catalytic electron transfer reaction between water and $Ce^{4+}$ cations. The reaction, while thermodynamically feasible, was very slow in the absence of a catalyst. Note the division by $\frac{1}{2}$ in front of the water electron transfer, which occurred because 1.23 V was for one electron. The fate of $Ce^{3+}$ was studied here as it was deposited on the catalyst surface at the reaction conditions.

Ru-based catalysts have remained the focus of intense efforts since the discovery of the Ru oxo-bridge dimer activity for water oxidation in 1982 [8]. The interest in Ir-based catalysts surfaced decades later after the first publication about bis-phenylpyridine Ir catalysts activity for OER in 2008 [9]. Despite their scarcity and high cost, $IrO_2$ and $RuO_2$ are among the most active commercial anodes used in water electrolysis since the 1970s [10,11]. Both metal oxides sit at the top of the volcano plot, and have optimal oxygen (O) adsorption energy, which was postulated to be one of the important criteria for OER activity [4,12–14]. Currently, Cp* (pentamethylcyclopentadienyl anion)-based homogeneous catalysts are commonly studied and were reported to be the most active, reaching maximum turnover frequencies (TOF) of ca. 20–40 $min^{-1}$ [15–17]. However, these catalysts decompose or take different chemical forms during the oxidation reaction [15]. In addition, the generation of unknown insoluble nanoparticles during the $Ce^{4+}$ reduction reaction when using $[Ir^{3+}(Cp*)][4,4'-(OH)_2-2,2'-bipyridine]cat(H_2O)]^{2+}$ was reported to be partly due to the oxidation of its ligands [18]. As indicated by others [19], because oxides of Ru and Ir are efficient catalysts for water oxidation (in electrocatalytic systems), considerable doubts exist as to whether the initial metal complex is the true catalytic species, and resolving this homogeneity-heterogeneity issue is vital to gain fundamental knowledge of the catalytic mechanism. Therefore, the breakdown of the ligand complex structure of the previously mentioned molecular iridium catalysts, through the use of CAN as a sacrificial agent, hinders the ability to study active sites.

Turnover frequency (TOF), a term initially introduced by M. Boudart into heterogeneous catalysis [20,21], has been widely used to gauge biochemical reaction kinetics; the most common example is likely that based on Michaelis–Menten kinetics [22]. While the definition of TOF is deceptively simple—the number of molecules (products) made in a unit of time over the number of sites used inside a reactor (volume)—the main issue is obviously the lack of understanding of what is meant by a reaction site on a solid particle, and how it can be analyzed and computed. On a binary metal oxide single crystal in UHV conditions, the problem is often simpler since one, in principle, can count the number of a given site before and after adsorption, and then using STM after the reaction, for example, as in the case of gold clusters on $TiO_2$ (110) single crystal [23]. For powder materials, the situation is more complex. It was shown that a well-defined powder system, such as ZSM-5 titration of Brønsted sites, was successful for TOF calculations for hydrocarbon cracking [24,25] (and references therein). However, when dealing with a metal supported on an oxide powder (in particular), questions related to the effects of particle size/structure and the interface interaction with the support surface become daunting. The further complexity in this work is the use of ions, $Ce^{4+}$, that are reduced to $Ce^{3+}$, and the possible precipitation (or irreversible adsorption) of both during the reaction. While some approximations need to be conducted in order to normalize reaction rates, there is a risk of unnecessary reductionism. This study aims to probe into a system composed of supported Ir clusters upon their reaction with $Ce^{4+}$ cations in aqueous environment for water oxidation to $O_2$. The support $TiO_2$ anatase was chosen as it is among the most understood and metastable metal oxides and has been shown to effectively disperse $IrO_2$ using simple synthetic techniques (e.g., wet impregnation and incipient wetness) [26–30]. It is also a reducible metal oxide that has shown considerable metal support interaction properties, and this may lead to certain specific interactions at the $IrO_2/TiO_2$ interface. This is particularly important because the catalysts are calcined at relatively high temperatures and, therefore, the interaction with the support may differ depending on the amount of Ir present (and therefore, the particle size distribution).

In order to extract reliable reaction rates for water oxidation, a stable Ir-based catalyst was synthesized and studied. The objective was to link the metal size and dispersion to the reaction rate in order to probe into the reaction kinetics. Transmission electron microscopy (TEM), scanning transmission electron microscopy (STEM) and X-ray diffraction (XRD) were used to investigate the morphological properties of the catalysts, while X-ray photoelectron spectroscopy (XPS) was used to identify the chemical state and atomic percentage of Ir and Ce cations on top of $TiO_2$. Temperature-programmed reduction (TPR) was used to evaluate and compare the reducibility and heterogeneity of the $IrO_2$ particles and a wide range of Ir contents in order to extract quantitative data as well as to probe into the $IrO_2$ amorphous and crystalline structures on top of $TiO_2$. The catalytic reaction was conducted in the presence of $Ce^{4+}$ cations at different concentrations, from which the reaction rates were extracted. $O_2$ production was measured using GC-TCD and, in a selected set of catalysts, UV-Vis was also collected to monitor the formation of $Ce^{3+}$ cations (at 327 nm). Particular attention was given to rate normalization (TOF) based on spectroscopic (XPS), microscopic (TEM) and kinetic (TPR) methods.

## 2. Results and Discussion

### 2.1. Catalyst Dispersion and Structure

Figure 1 shows STEM images of $IrO_2/TiO_2$ (anatase) catalysts with different Ir wt.% values along with their corresponding $IrO_2$ clusters' size distributions. The as-prepared catalysts contained Ir atoms in their oxidized state, most likely as $IrO_2$ clusters (monitored by their Ir4f XPS lines, presented below). The Ir oxide particles appeared to be bright due to the relative difference in atomic mass when compared to Ti atoms of the $TiO_2$ support particles [31]. The mean particle size was calculated by measuring the diameter of a minimum of 100 clusters. The relationship between particle size and geometry is complex, particularly because it is a function of their structure. While there is less work

focused on linking the number of atoms of metal oxide clusters to their size and structure, computationally, one can extract information from the wealth of studies on metal clusters. For example, Ir clusters were studied using DFT/PW-91 with Ir $5d^8$ and $6s^1$ treated as the valence electrons in the work of Chen et al. [32], in which their geometric structure was computed as a function of the number of atoms extending from 2 to 10. Except $Ir_2$, these were triangular and square-based structures. The clusters' size, extracted from the 2D distance of their base (from Figure 2 of [32]), was found to be between 0.22 and 0.34 nm for the 3- to 10-atom clusters. It is likely that the most studied clusters' size and associated number of atoms of an FCC element are those of Au, and this provides a guide for larger particle sizes. For example, $Au_{55}$ has a size of 1.44 nm, while the size of $Au_{13}$ is 0.86 nm [33]. Because of volume expansion when compared to metals it is likely that, for all catalysts studied in this work, the total number of Ir atoms per cluster did not exceed 15 (calculated in the table inside Figure S1 from the volume, density and atomic mass of $IrO_2$ assuming a spherical shape). It should be noted that the diagonal of the 3D-tetragonal structure of the rutile $IrO_2$ (the unit structure contains two full Ir atoms and nine in total) is about 0.7 nm. The mean particle size increased logarithmically with the increasing of the wt.% of $IrO_2$ on $TiO_2$ anatase, while the dispersion computed from the mean particle size was anti-symmetric to the increase in the mean particle size.

Additional TEM, SAED, EDX, and STEM results of the $IrO_2/TiO_2$ are presented in Figure 2. The TEM image shows distinct sub-nanometer dark spots on ca. 20 nm $TiO_2$ particles. The SAED image of a catalyst area of ca 0.2 $\mu m^2$ shows diffraction patterns corresponding to (101), (200), and (004) crystallographic planes for a typical $TiO_2$ anatase support. EDX analysis from the image area indicated that the darker spots are Ir-containing clusters. As can also be seen from STEM images in Figure 2, a non-negligible number of clusters seem to be in 2D structure due to their small size. Based on contrast observations, some $Ir^{4+}$ cations may have substituted $Ti^{4+}$ cations of the support (probably during the calcination process at 400 °C)—see Figure 2d in particular. The effect of reduction on this will be discussed later in the Temperature Programed Reduction section.

XPS was used to monitor the core level of Ir 4f on the $TiO_2$ surface. The XPS Ir4f of Ir and its oxides was previously studied by numerous researchers [34,35]. The XPS $Ir4f_{7/2}$ of metallic Ir is very close to 60.0 eV with a spin orbit splitting of ca. 3.0 eV. These have an asymmetric shape, which is typical of metallic states. The main oxidation state of Ir is to the +4 state, giving rise to $Ir4f_{7/2}$ at about 62 eV, also with a spin orbit splitting of ca. 3.0 eV. Because of changes in exact position and line shape due to the nature of the material (amorphous, crystalline powder, and single crystal) and instrument resolutions, we opted to study the XPS of an as-received $IrO_2$ compound that we used for catalyst preparation. It was heated in the UHV chamber at incremental temperatures where a partial conversion to metallic Ir took place (Figure S2), as outlined next.

The spectra of $IrO_2$ powder are presented as collected without further calibration, with low electrical resistivity (10–100 $\mu\Omega$ cm), in the form of a thin pellet fixed onto a Ta plate that was heated from the back by an electron beam heater. (10–100 $\mu\Omega$ cm), in the form of a thin pellet fixed onto a Ta plate that was heated from the back by an electron beam heater. The O1s at 530.0 eV was typical of that of an oxide with considerable presence of irreversibly adsorbed water at about 532.5 eV (indicated by a line in the spectra); the surface hydroxyls signal would fall somewhere in between, at about 531.5 eV. The heating of $IrO_2$ did not cause a significant shift in the overall signal, with the only noticeable change being the desorption of water from its molecularly and dissociatively adsorbed state. At 970 K, there were still non-negligible amounts of O1s (about half of the original amount), indicating that only part of $IrO_2$ was reduced (Figure S2). The XPS Ir4f region of as-received $IrO_2$ was fitted as two main peaks and their satellites. The presence of an XPS $Ir4f_{7/2}$ at about 62 eV and the quasi absence of a signal at ca. 60 eV is a signature of $Ir^{4+}$ cations in $IrO_2$. Heating to 620 K resulted in a slight shifting of the signal to lower binding energy, together with a collapse of both spin orbit split peaks. This was due to a partial reduction to Ir metal, resulting in an XPS $Ir4f_{5/2}$ of $Ir^0$, which would fall between the two-spin orbit-split

Ir4f peaks of $Ir^{4+}$. We opted not to curve-fit this spectrum because of the uncertainty of the exact reduced states and their contributions, as well as the multiple satellites, as these are not within the scope of the work. Further heating to 770 K shifted the peaks, with the highest signal appearing to originate from that of $Ir^0$. At 970 K, the signal was fitted for $Ir^0$, while the dashed lines represent the signal from other contributions, including satellites of $Ir^0$ as well as the remaining $Ir^{4+}$ cations of $IrO_2$. It is worth noting the considerable curvature of the high binding energy of the region at about 66 eV (indicated by an arrow in the figure), which is an indication of the decreased contribution of the $4f_{5/2}$ of $Ir^{4+}$ cations to the overall signal.

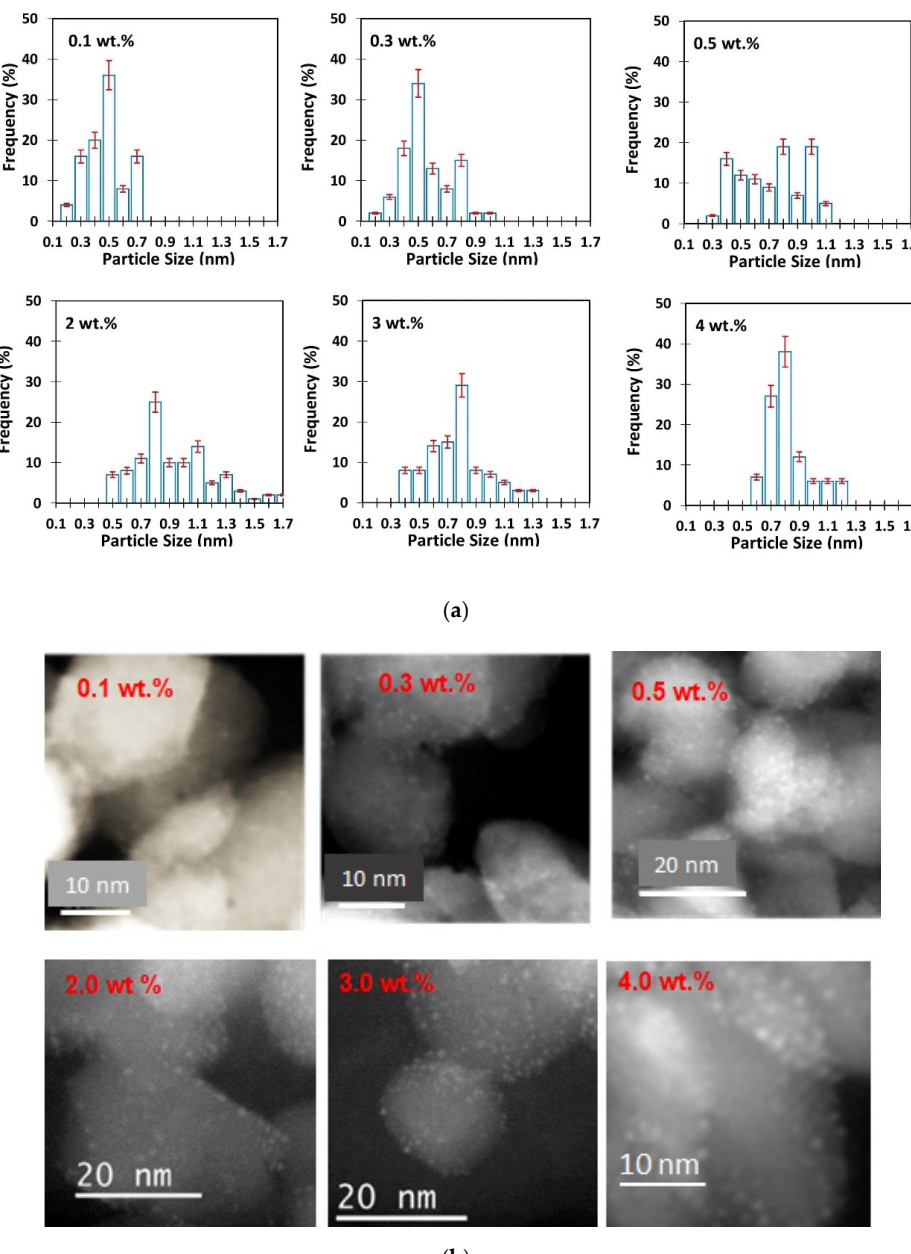

**Figure 1.** (**a**) $IrO_2$ particle size distribution from (**b**). (**b**) STEM images of a selected number of $IrO_2/TiO_2$ catalysts with the indicated Ir wt.% loading. $IrO_2$ clusters are shown as white dots on top of the anatase $TiO_2$ support (grey).

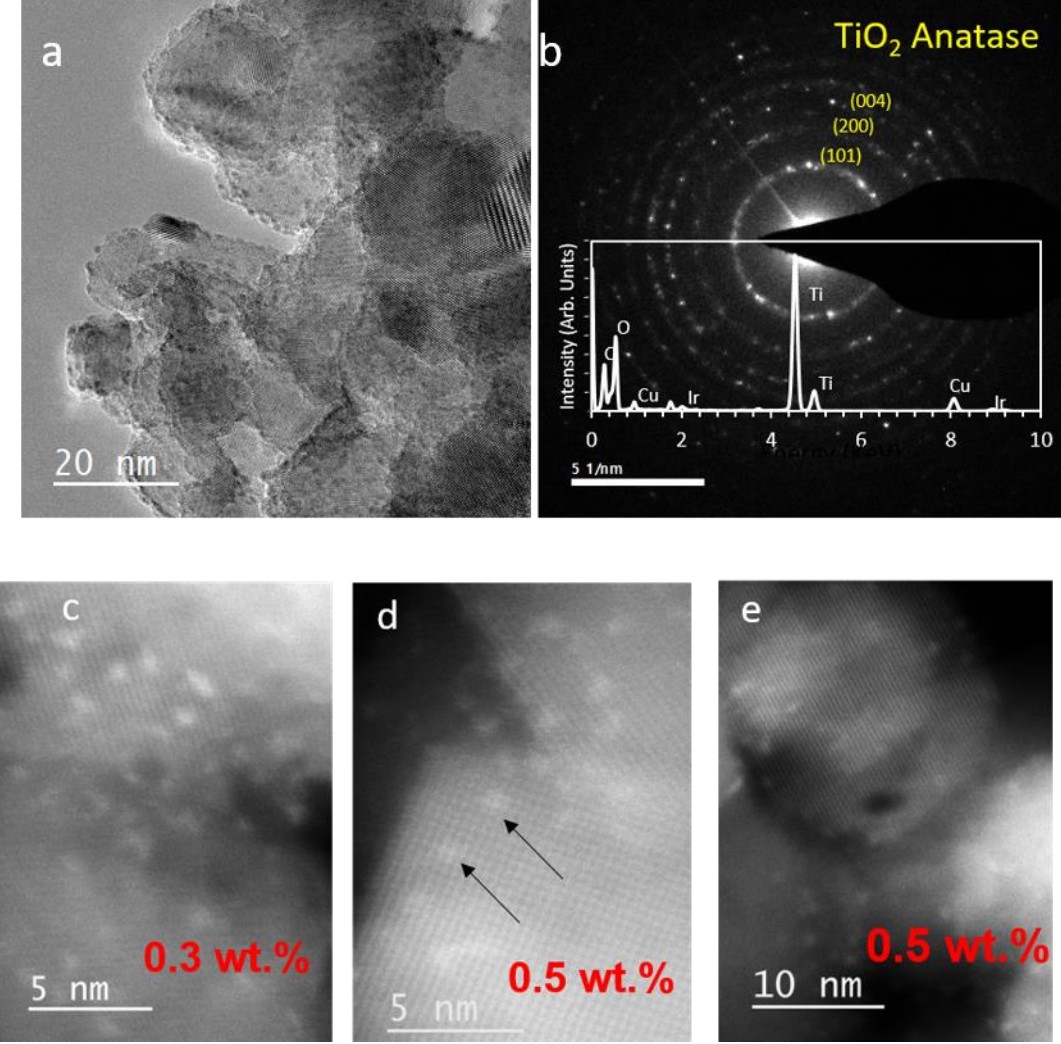

**Figure 2.** Bright and dark field TEM images of IrO₂/TiO₂ catalysts. (**a**) TEM image of the 4 wt.% IrO₂/TiO₂; (**b**) a selected area electron diffraction (SAED) pattern of ca 0.2 μm²; inset: respective EDX analysis in which Ir is shown; the Cu EDX signal is from the grid. The SAED image of a catalyst area shows diffraction patterns corresponding to the (101), (200), and (004) crystallographic planes for the TiO₂ anatase support. (**c–e**) HRSTEM images for 0.3 and 0.5 wt.% Ir on TiO₂ anatase at different magnifications. The arrows in (**d**) point to a possible Ir⁴⁺ substitution of Ti⁴⁺ cations in the TiO₂ anatase lattice.

The XPS of Ir4f, when deposited on TiO₂, showed a further complication. There was some contribution of Ti3s (at about 61.2 eV, therefore, overlapping with Ir4f) in this region, which needed to be subtracted to obtain quantitative information, particularly for low Ir loading. To achieve this, the Ti3s of a pure TiO₂ was monitored, then the peak area ratio of the Ti3s/Ti2p was computed. The ratio was found to be equal to 1/14. Figure 3 shows the XPS Ir4f spectra for 4 and 1 wt.% IrO₂/TiO₂ before and after Ti3s peak subtraction. The peak-fitting parameters for both catalysts were taken from the pure IrO₂ reference sample, shown in Figure S2, indicating that Ir exists as IrO₂ alone; no noticeable beam damage was seen during XPS data collection. For both catalysts, the XPS Ir4f spectrum was composed of two peak doublets; the doublet with peaks at 61.8 eV (FWHM = 1.9 eV) and 64.7 eV (FWHM = 2.0 eV) assigned to the Ir4f $_{7/2}$ and Ir4f $_{5/2}$ orbitals, respectively, and the other wider doublet with peak positions at 62.3 and 65.2 eV, assigned to their satellites. Based on these results, it is clear that the as-prepared catalysts were composed predominantly of IrO₂ clusters (and not Ir metals) on top of TiO₂ (anatase) and, therefore, the clusters seen by HRTEM were those of IrO₂. Figure S3 and Tables S1 and S2 give more information on the atomic composition and peak-fitting parameters.

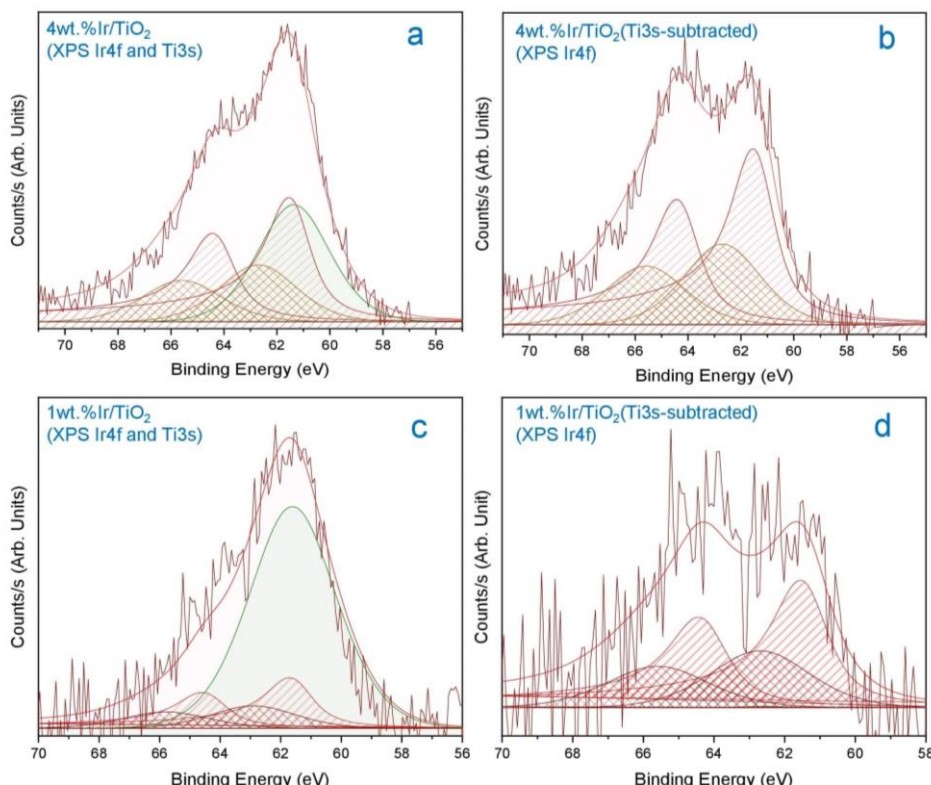

**Figure 3.** XPS spectra of the Ir 4f region of 4 wt.% $IrO_2$/$TiO_2$ (anatase) (**a**) before and (**b**) after Ti 3s subtraction. XPS spectra of the Ir 4f region of 1 wt.% $IrO_2$/$TiO_2$ (anatase) (**c**) before and (**d**) after Ti 3s subtraction (the Ti3s peak is shown in green in each).

To further analyze the catalysts, XRD was conducted. This is particularly important because $IrO_2$ has a rutile structure with the 2θ line of its (110) plane very close to that of the rutile $TiO_2$ (if present). Figure 4a shows XRD patterns of a commercially available $TiO_2$—(anatase 88 wt.%-rutile 12 wt.%, Aldrich)) and the as-prepared $IrO_2$/$TiO_2$– anatase catalysts. For the as-prepared catalysts, the peaks at 2θ of 25.3°, 37.8° and 48.1° correspond to $TiO_2$ anatase (110), (004) and (200) planes, which is in agreement with SAED patterns of the 4 wt.% $IrO_2$/$TiO_2$ catalyst [36]. The commercial sample shows peaks for both $TiO_2$ phases. Apart from peaks at ca. 28.0° and 34.7°, no noticeable difference is observed in the XRD patterns as a function of Ir wt.% loading. Because the conversion of a small percentage of $TiO_2$ anatase to rutile upon deposition of sub-nanometer particles of $IrO_2$ was reported [37], there is some doubt as to whether the ca. 28.0° line is due to rutile $IrO_2$ or to some minor phase change of $TiO_2$ anatase during the catalyst preparation. We show, in the following arguments, that this ca. 28.0° line was indeed due to $IrO_2$ and not due to $TiO_2$ rutile: (i) There is a 2θ difference of 0.2–0.3° between $TiO_2$ rutile (110), at 27.6°, peak and $IrO_2$ (110) peak at 28.0°. (ii) The peaks at 2θ values of 28° (110) and 34.7° (101) increased in intensity with the increasing of the Ir content (inset in a). (iii) To further study this, we heated the 4 wt.% $IrO_2$/$TiO_2$ to higher temperatures and compared it to the same $TiO_2$ without $IrO_2$ at the same temperatures (Figure 4b). An increase in intensity of these two peaks, due to the improved crystallinity of $IrO_2$ with temperature, was noticed. The heated $TiO_2$ anatase alone, in relation to temperature, did not show these peaks. (iv) It is also worth noting that we did not detect rutile $TiO_2$ particles via TEM. Moreover, the relationship between these two diffraction peaks and the reduction of Ir during TPR, together with the TEM images, are presented in the TPR section below.

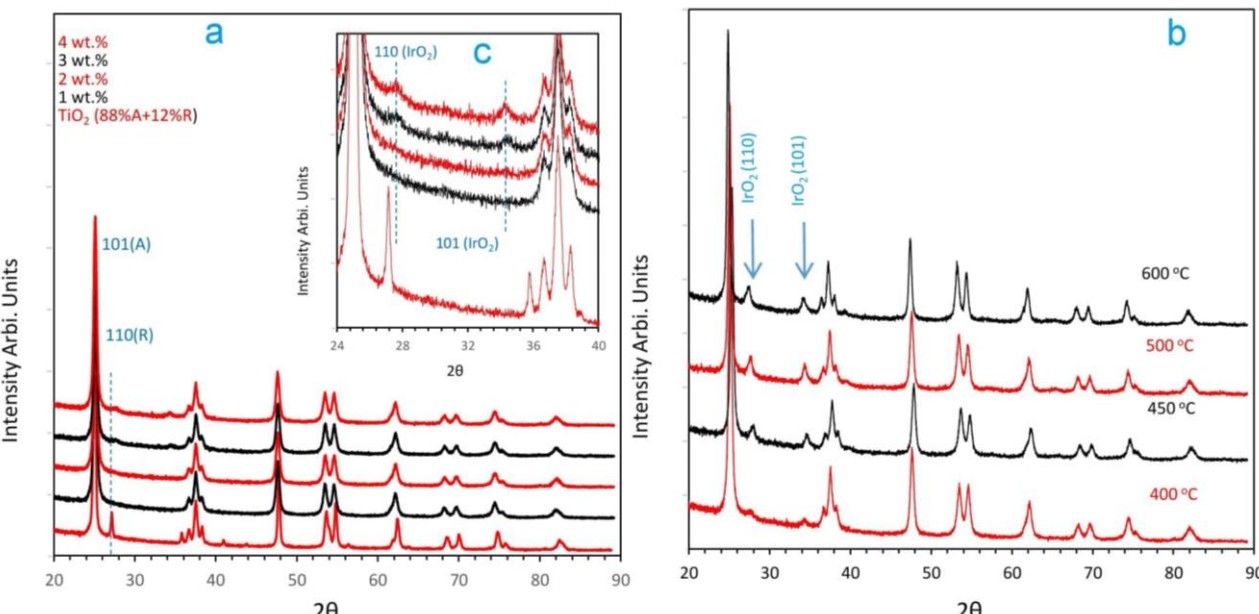

**Figure 4.** XRD patterns of IrO$_2$/TiO$_2$ catalysts (**a**) with the indicated Ir wt.% loading and (**b**) with 4 wt.% Ir on TiO$_2$ as a function of calcination temperature. (**c**) The inset in (**a**) is an expanded section to highlight the (110) and (101) peaks attributed to IrO$_2$ as a function of their concentration. The TiO$_2$ (88% A + 12% R) is shown as a reference and was not used as a support in this study.

### 2.2. Temperature-Programmed Reduction and Effect of Calcination Temperature on Catalyst Dispersion

Figure 5 presents the TPR results for IrO$_2$/TiO$_2$ catalysts. All heating rates were conducted at 10 °C/min at a flow rate of 50 mL min$^{-1}$. TPR profiles, while bring quantitative results (hydrogen uptake), are qualitatively difficult to understand for metal-supported systems, largely because they are affected by different solid properties such as particle size, loading %, and increased metal diffusion with temperature, which is in part linked to the nature of the interaction with the support. In general, the oxides of noble metals such as Pt, Pd, and Rh, dispersed on a support with particles in the nanometer size, are reduced below 150 °C. Larger particles tend to be more crystalline and are, therefore, reduced at temperatures close to those of the bulk oxide. In addition, a given support (if reducible) such as TiO$_2$ or CeO$_2$ can be reduced by hydrogen atoms from the metal surface at still higher temperatures (spillover effect).

Thus, one expects, in the case of Ir/TiO$_2$, at least three regions, each of which would be affected by the metal loading, the particle size and the degree of crystallinity. In Figure 5, we highlighted two regions that largely separate the two systems: the so called "amorphous" phase (dispersed particles of 1 nm or so in size) and crystalline phase that were not present in the fresh catalyst, but were made during the reduction process, in which we additionally inferred the role of the TiO$_2$ support. The figure presents TPR profiles ranging from 0.1 wt.% to 4 wt.% of IrO$_2$. Most of these were studied by TEM (some are presented in Figures 1 and 2).

The first observation is that of the increase in the TPR signal in both regions with the increasing of the IrO$_2$ wt.%. The increase is, however, not similar for both regions. Second, it seems that the first region is largely composed of one peak when IrO$_2$ has a high density (at 1.5 wt.% and above) on the surface, while the second region shows multiple clear and pronounced peaks. It is likely that a non-negligible part of IrO$_2$ sinters upon heating rather than being reduced. Statistically, the competition between surface diffusion of IrO$_2$ on TiO$_2$ (anatase) and its reduction to Ir metal is in favor of the former at high surface densities, and of the latter at low surface densities. Figure 5b,c show the particle size distribution obtained from STEM images of the IrO$_2$ of an as-prepared 1 wt.% IrO$_2$/TiO$_2$, and the same

after TPR that was stopped at 120 °C. The objective was to see changes that may have occurred during the reduction in the first step. The mean particle size increased by about 0.2 nm after being heated to 120 °C. Since the reduction of a metal oxide to a metal is accompanied by a decrease (not an increase) in the particle size (the volume contraction of $IrO_2$ to Ir is equal 55%; the unit volume of the tetragonal $IrO_2$ is 65.56 $Å^3$ (containing two Ir atoms) and that of the fcc Ir metal is 57.9 $Å^3$ (containing four Ir atoms)), the observed increase in particle size must be related to non-negligible sintering (volume contraction based on density gives a similar number: the $IrO_2$ density is 11.66 $g/cm^3$ while the Ir is 22.56 $g/cm^3$). It is, however, not possible that all $IrO_2$ clusters were reduced to Ir metal at this temperature (otherwise, there would have been no reduction peaks left in region 2), and, since we did not see larger particles, one therefore questions the attribution of the low temperature peak solely to amorphous $IrO_2$; it might have been related to a partial reduction in the clusters, and the process may have also been driven by the kinetics of cluster diffusion in addition to those of the reduction of an oxide.

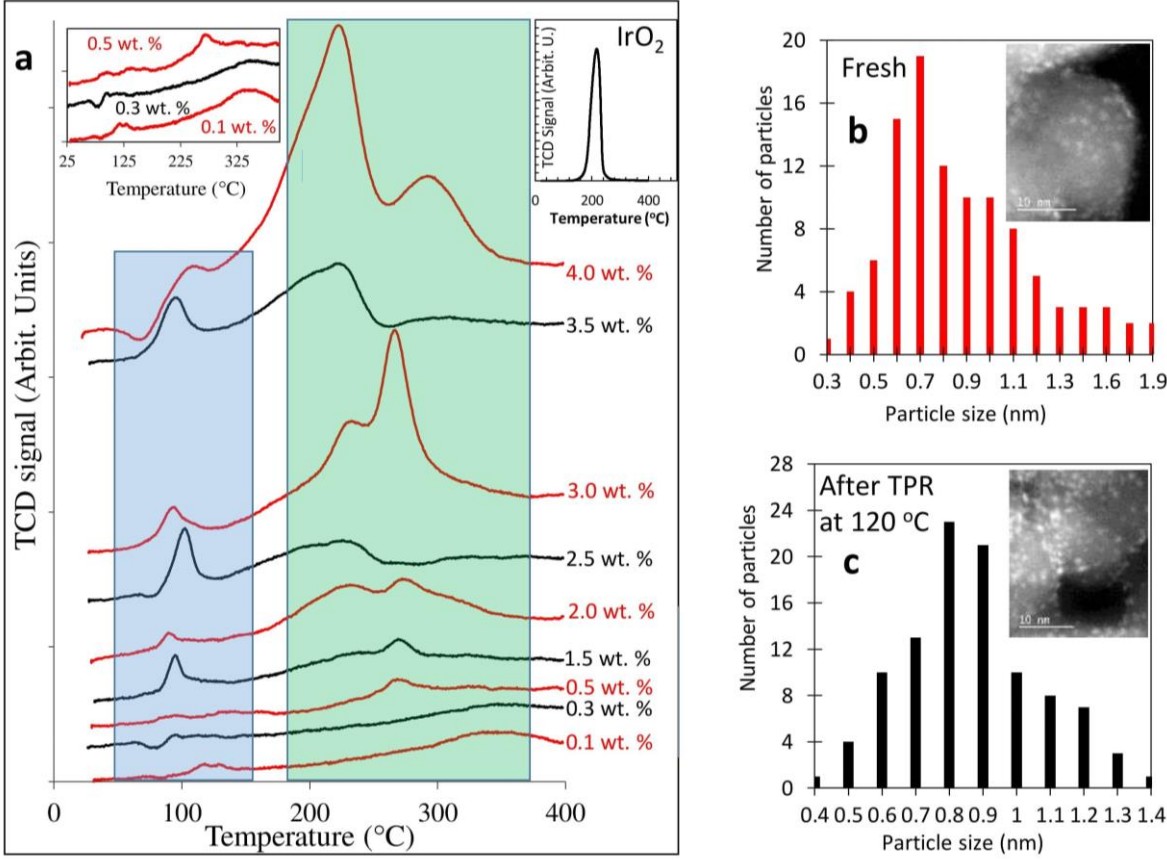

**Figure 5.** (**a**) TPR profiles of $IrO_2/TiO_2$ catalysts with the indicated Ir wt.%: 0.1 wt.% (59.5 mg); 0.3 wt.% (58.4 mg); 0.5 wt.% (59.4 mg); 1.5 wt.% (61.3 mg); 2.0 wt.% (62.2 mg); 2.5 wt.% (61.7 mg); 3.0 wt.% (63.3 mg); 3.5 wt.% (63.1 mg); 4.0 wt.% (100 mg). The top inset in (**a**) is a magnification of the signal in region 1 for the 0.1–0.5 wt.% $IrO_2$ catalysts, while the top right inset is the TPR of polycrystalline $IrO_2$. (**b**) Particle size distribution obtained from STEM images of 1 wt.% $IrO_2/TiO_2$ catalyst before (**b**) and after (**c**) TPR up to 120 °C. The mean particle size increased from 0.6–0.7 nm to 0.8–0.9 nm. The insets in (**b**,**c**) are representative STEM images where some sintering can be observed (reduction without sintering would result in a decrease in the particle size by about half (see text for further explanation)). The first highlighted region is largely attributed to amorphous phases, while the second one is attributed to crystalline phases of $IrO_2$ (see text for more interpretations).

The second region is equally complex. It is composed of two peaks for most catalysts studied in the series. The position and relative contribution of these two peaks oscillates with the increasing of the $IrO_2$ weight %. This is most likely dominated by bulk reduction,

although some reduction of $TiO_2$ might also occur. To further study this region in particular, we conducted different TPR experiments on $IrO_2$ that was calcined at different temperatures from 400 to 600 °C.

These are presented in Figure 6a. We first note that the defined peak in region 1 gradually disappears with increasing calcination temperatures. This is also associated with the increasing symmetry of the main TPR peak. This can be noticed by the area framed in blue lines that decreases with increasing calcination temperatures. This again points to the non-homogenous nature of the reduction, where $IrO_2$ clusters diffuse (and sinter) during the reduction at 400 °C. The increasing of the calcination temperature, prior to reduction, resulted in the sintering of a larger fraction, which, in turn, allowed the reduction process to occur in a narrower temperature window. One can then monitor this by following the XRD signal of $IrO_2$ (from Figure 4). The more crystalline the $IrO_2$, the larger the signal. The XRD data in Figure 6b are not quantitatively calibrated but they are used as a guide for the formation of the crystalline phase. It is to be noted that they are not normalized to the anatase $TiO_2$ peak area either, since the latter was found to also increase in crystallinity with the increasing of the calcination temperature (the FWHM of the anatase 101 peak decreased from 0.448 to 0.378 and shifted from 25.34° to 25.03° upon heating from 400 °C to 600 °C). The two high temperature peaks observed for the catalyst that was calcined at 400 °C collapsed in one larger peak together with a shift to higher temperatures. This was most likely the result of a particle size effect. Figure 7 presents STEM images of $IrO_2/TiO_2$ that was calcined at temperatures between 300 and 600 °C. While we have not conducted TPR experiments for the samples calcined at 300 °C, they are included to make the distinction clearer. It is clear from the images that sintering occurred even at 400 °C when compared to the 300 °C calcination temperature. Sintering became pronounced at 500 °C where $IrO_2$ particles agglomerated together, thereby making larger ones, and was dominant at 600 °C.

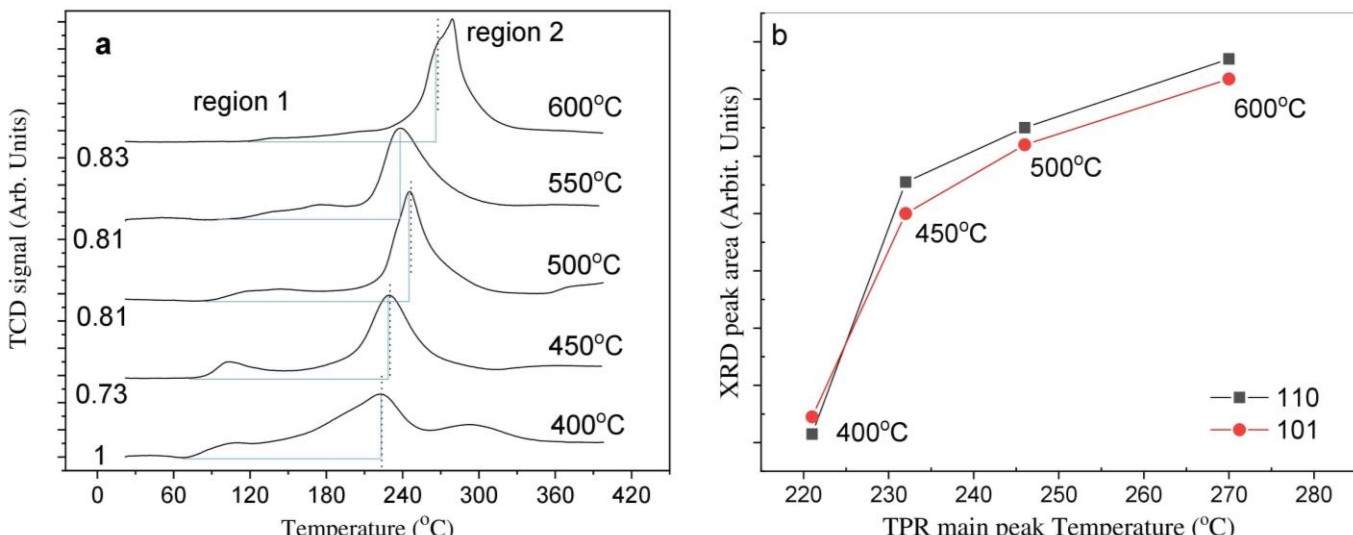

**Figure 6.** (**a**) TPR of 4 wt.% $IrO_2/TiO_2$ calcined at the indicated temperatures. The numbers on the left-hand side are the relative peak areas obtained from the TCD signal. (**b**) A plot of the XRD peak areas of the (110) and (101) lines of $IrO_2$ as a function of the TPR peak temperature at the different calcination temperatures (the dashed lines in (**a**)). TPR was conducted at a ramping rate of 10 °C/min and a flow rate of 50 mL min$^{-1}$ (10% $H_2$ in Argon).

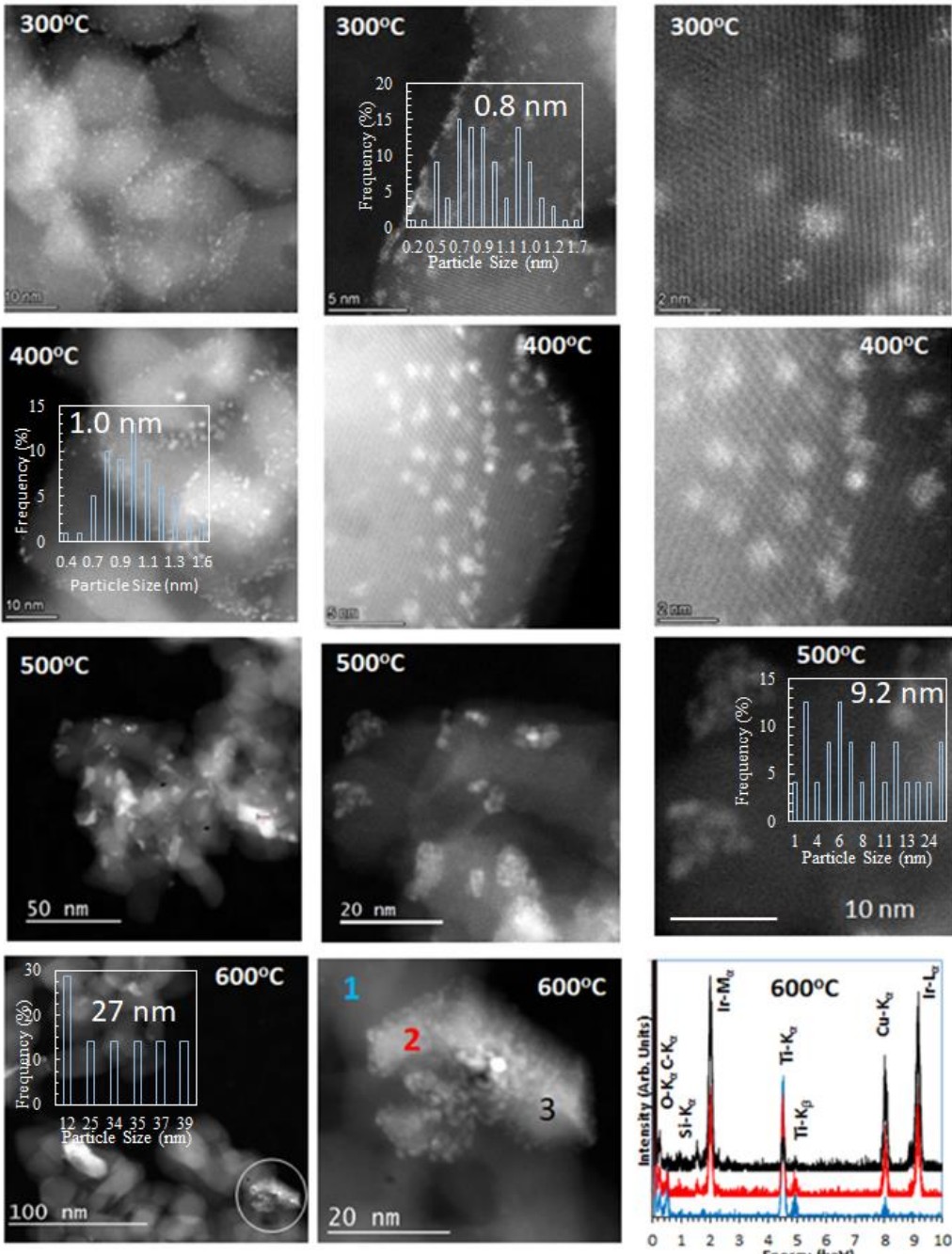

**Figure 7.** STEM images of 4 wt.% $IrO_2/TiO_2$ at different magnifications for samples calcined between 300 and 600 °C for four hours each. Clustering became dominant at 500 °C. The mean particle size is given in the insets. It increased from 0.8 nm upon calcination at 300 °C to 27 nm when the calcination temperature was raised to 600 °C.

Quantitative analysis of the complete series is presented in Figure 8. The corresponding individual peaks with peak integrations are given in Figure S4. The straight solid line at $H_2/IrO_2 = 2$ indicates the expected stoichiometric ratio if all $IrO_2$ has been reduced to Ir and there is no to $H_2$ consumption caused by $TiO_2$ reduction. For the series of catalysts between 0.3 wt.% and 4 wt.%, the total amount of hydrogen oscillates around this value. The large deviation for the 0.1 wt.% might be due to large errors, because of the high background compared to the signal, but it could also be due to a strong interaction between atomically dispersed Ir cations on (or in) $TiO_2$, where their reduction would also cause further reduction in the support.

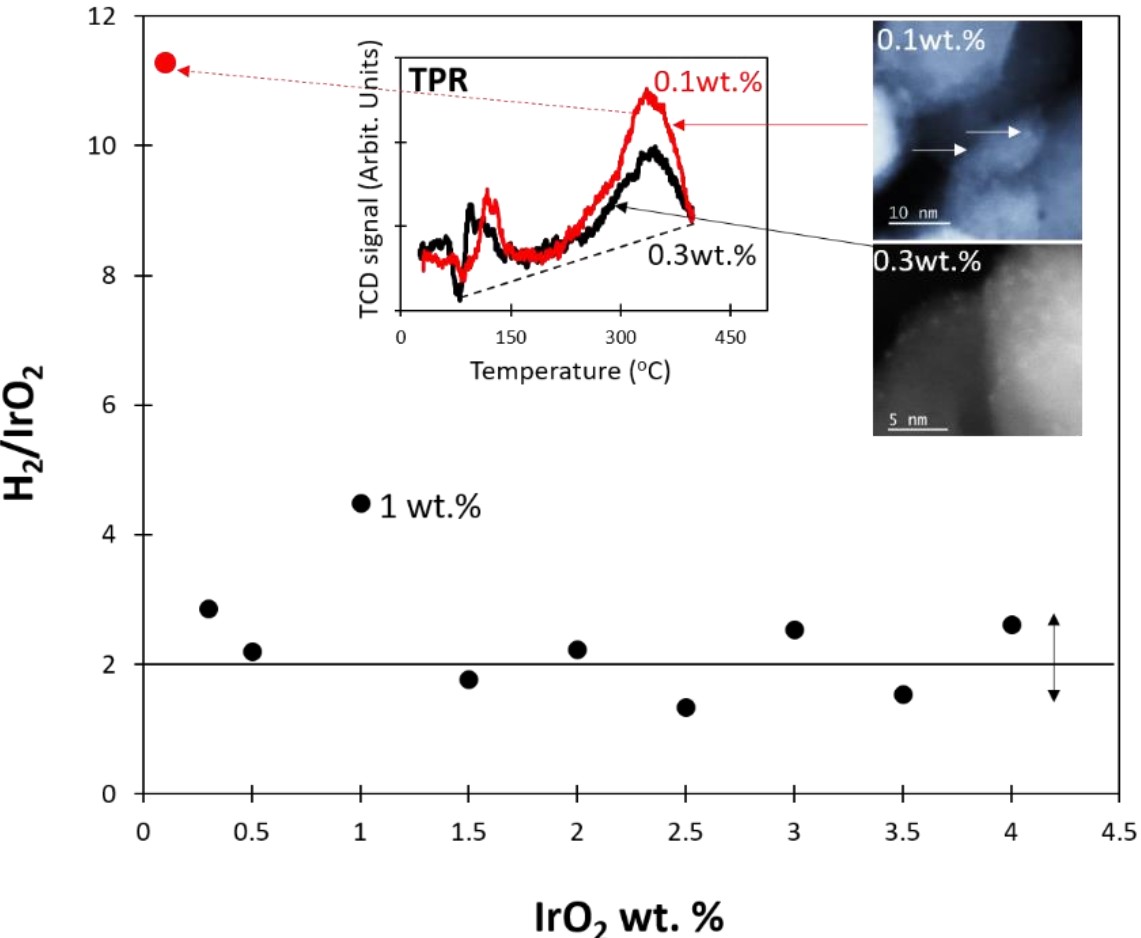

**Figure 8.** Quantitative analysis of the TPR series of $IrO_2/TiO_2$ as a function of Ir wt.%. The Y-axis presents the $H_2/IrO_2$ ratio. The dashed line indicates the expected ratio for a full reduction. The 0.1 wt.% profile is not included as quantification was not possible due to the low coverage. The 1 wt.% profile is also not included as it showed high deviation for the series (the $H_2$ to Ir ratio is found to be about 4). The particle size distribution and size of $IrO_2$ of these catalysts are given in Figures 1, 2 and 7.

### 2.3. Oxygen Production from Water in the Presence of $Ce^{4+}$ Cations

In the following, we present and discuss the OER results; these were obtained by monitoring $Ce^{3+}$ cations (one of the reaction products) using UV-Vis spectroscopy and monitoring $O_2$ production (the other reaction product) using gas chromatography. In conducting this, we mainly studied two parameters: The effect of $IrO_2$ loading and the effect of $Ce^{4+}$ cation concentrations on the reaction rate.

### 2.3.1. Effect of $[Ce^{4+}]$

Figure 9a presents $O_2$ production as a function of time with one catalyst (1 wt.% $IrO_2/TiO_2$) at different $Ce^{4+}$ concentrations. $O_2$ production increased with increasing $Ce^{4+}$ concentrations and the saturation was due to their complete consumptions, as discussed in greater detail below. The reaction could also be monitored, although not quantitatively using UV-Vis absorbance measurement, as shown in Figure 9b. $Ce^{3+}$ and $Ce^{4+}$ were found to absorb at 265, and 327 nm, respectively [17]. A gradual increase in absorbance due to the formation of $Ce^{3+}$ as a function of time evidences the progress of OER according to Equation (1). The initial intense orange color of $[Ce^{4+}]$ (0.182 M) aqueous solution faded to yellow over time, and then the solution finally became colorless. This occurred because the $Ce^{4+}/Ce^{3+}$ redox potential (ca. +1.7 eV vs. NHE) was lower than that of $H_2O/O_2$ and the reaction rate was catalytically enhanced in the presence of $IrO_2$. The decay of $Ce^{4+}$ cations

could not be monitored because $NO_3^-$ anions absorbed in the same region, between 320 and 340 nm [38]. For quantitative analysis, the change in concentration of $Ce^{4+}$ cations was computed from the equation $[Ce^{4+}]_o = [Ce^{3+}]_t + [Ce^{4+}]_t$, where $t$ is the reaction time (Figure 9c). The experimental ratio of $Ce^{4+}/O_2$ was calculated from the slope of the inset in Figure 9d. It was found to be equal to 3.72. $Ce^{3+}$ concentrations were calculated from the amount of $O_2$ produced as per Equation (1). While there was an increase in the reaction rate with the increasing of the initial concentration of Ce cations, as seen in Figure 9, the increase seemed to follow a Langmuir–Hinshelwood kinetic pattern, as shown in Figure 10d.

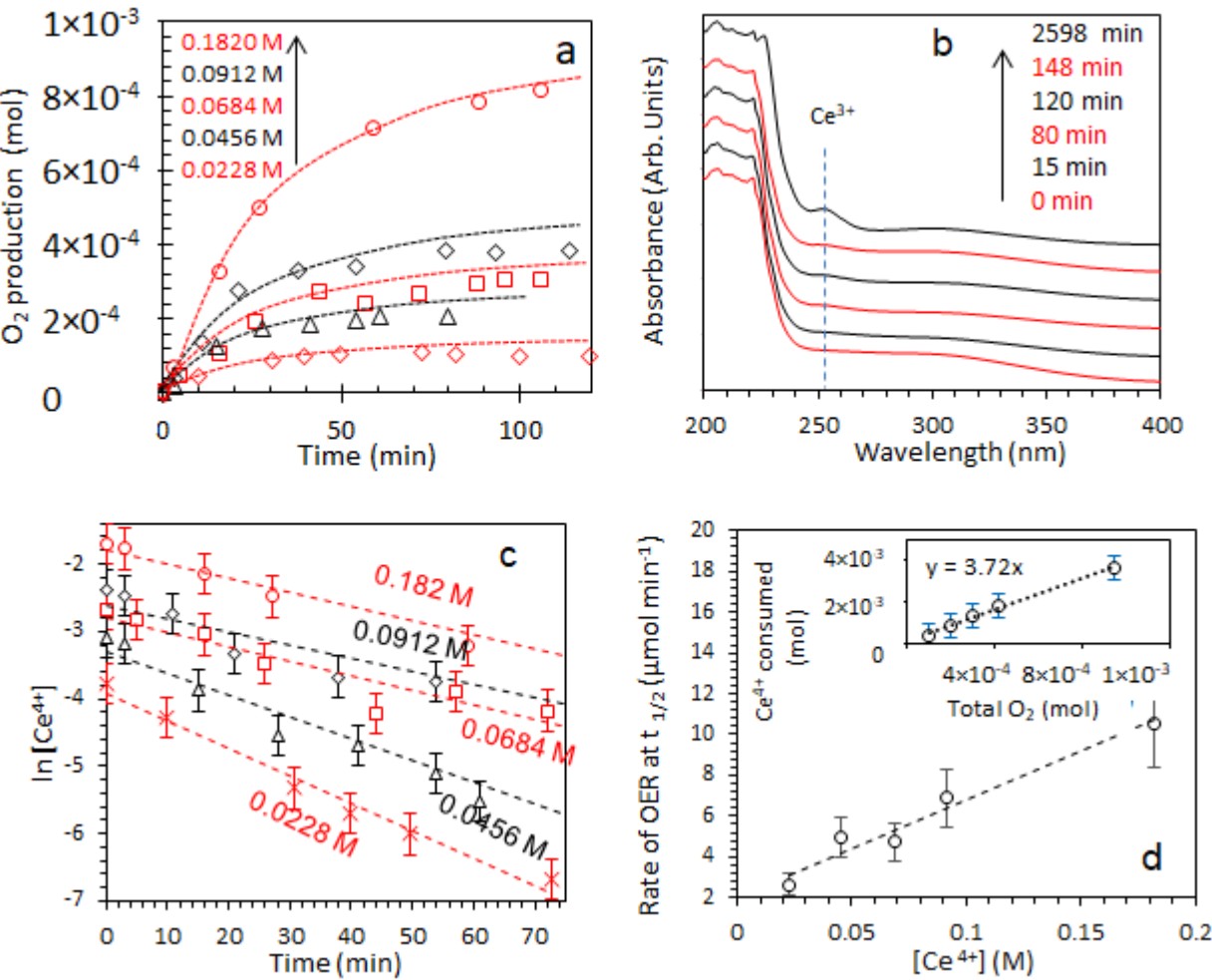

**Figure 9.** Reaction kinetics of water oxidation over 1 wt.% Ir on $TiO_2$ anatase at different reaction times and $[Ce^{4+}]$ concentrations. (**a**) Effect of $[Ce^{4+}]$ on $O_2$ production. (**b**) UV-Vis absorbance spectra monitoring the reduction of $Ce^{4+}$ to $Ce^{3+}$ during OER. (**c**) Computed $Ce^{4+}$ cation concentration (where $Ce^{4+}$ cations at time $t$, $[Ce^{4+}]_t$, is equal to $Ce^{4+}$ cations at time zero, $[C^{4+}]_0$, minus $Ce^{3+}$ cations at time $t$, $[Ce^{3+}]_t$) as a function of time. (**d**) Ln $[Ce^{4+}]$ as a function of time up to ca. 70 min. (**d**) Rate of OER as a function of $[Ce^{4+}]$ at time $t_{1/2}$ (where $t_{1/2}$ is the time at which 50% of $Ce^{4+}$ was consume); the inset presents a plot of the total moles of $Ce^{4+}$ cations as a function of the total moles of $O_2$ (a slope of 3.72 was obtained; the theoretical slope should be 4).

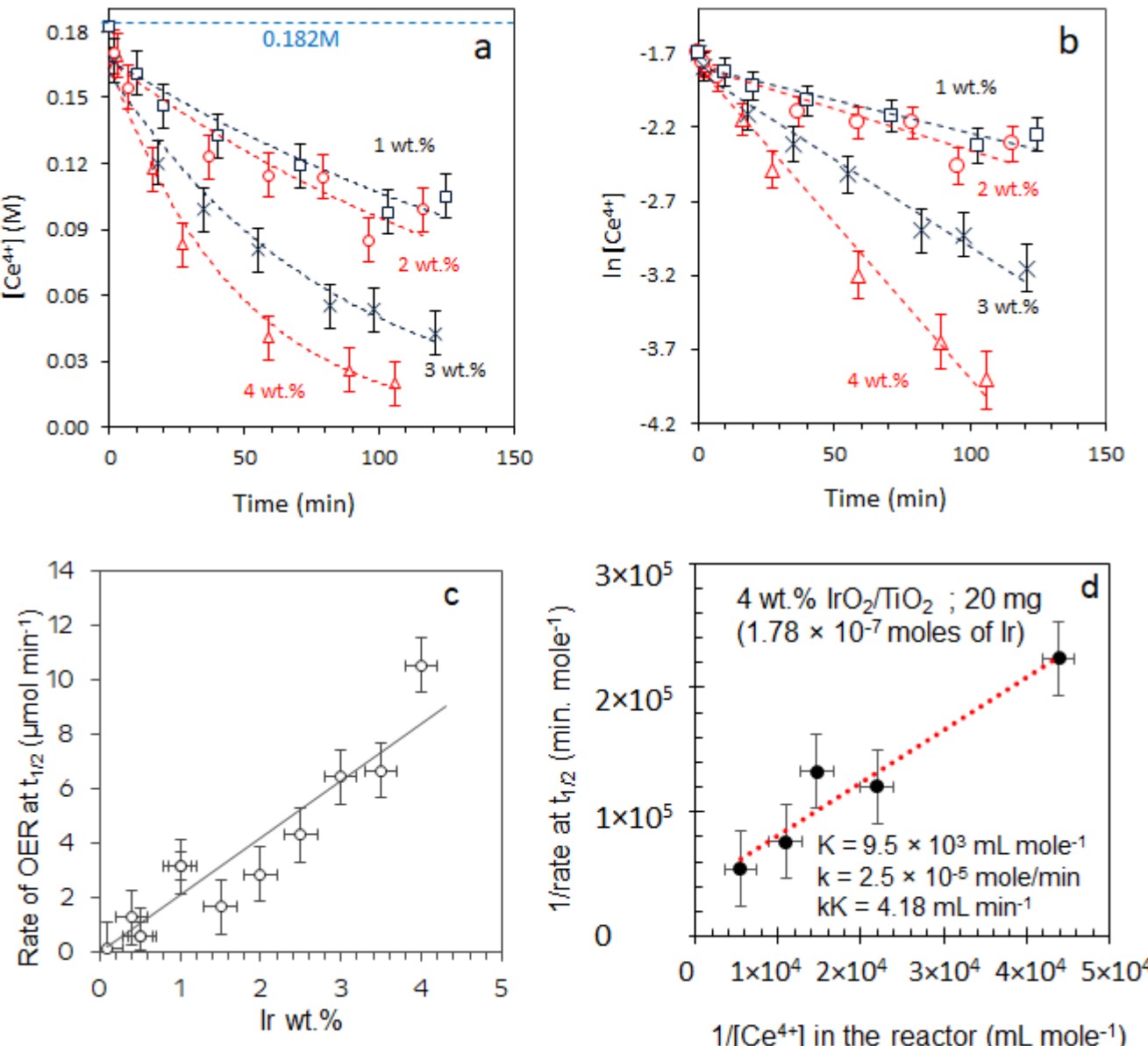

**Figure 10.** (**a**) Change of $Ce^{4+}$ cation concentrations with reaction time using different $IrO_2/TiO_2$ (anatase) catalysts with Ir wt.% values from 1 to 4. (**b**) Logarithm of the Y-axis in (**a**). (**c**) Reaction rates at $t_{1/2}$ as a function of Ir content of the different catalysts studied. (**d**) L-H plot of the data in Figure 10c. The two parallel red lines in (**c**) are for the confidence integral based on the results. All data were collected at the same condition with 0.182 M of initial $Ce^{4+}$ concentration.

### 2.3.2. Effect of $[IrO_2]$

Next, we present the reaction rates of the complete series at a fixed $Ce^{4+}$ initial concentration. Figure 10a,b shows that $[Ce^{4+}]$ changes logarithmically with time on all catalysts. There is an increase in the rate of $[Ce^{4+}]$ consumption with an increase in Ir loading. To further study the system, the rate of $O_2$ evolution, for a larger number of catalysts, was extracted, as indicated in Figure 10c.

The quasi linearity of the reaction rate indicated that dispersion was not a determining factor for catalyst activity up to the 4 wt.% of $IrO_2$, within the investigated range. Since electron abstraction from water (presumably in the form of adsorbed OH groups) to $Ce^{4+}$ cations was accelerated in the presence of $IrO_2$, the linear increase in reaction rate with the increasing of the latter was in favor of a bulk (total number of Ir atoms) mediated effect that was helped by the slow kinetics of the reaction (this point is further discussed in Section 2.5).

### 2.4. Catalyst Stability and the Location of Deposited Cerium Cations

Before proceeding to TOF calculations, we conducted a further study to monitor the stability of the catalyst over time. Figure 11 presents tests conducted on the 4 wt.% $IrO_2/TiO_2$ catalyst. The five runs gave the same amount of $O_2$, which saturates, within experimental errors, at the same reaction time. This means that $Ce^{4+}$ reduction to $Ce^{3+}$ and their possible deposition did not affect the Ir sites on $TiO_2$ for the reaction. At first sight, this might seem counterintuitive since Ce cations' deposition would eventually block or alter the catalytic sites. To further probe into this, we conducted STEM, HRTEM and XPS on the used catalyst. Our objective was to identify any structural variations in the $IrO_2/TiO_2$, as well as the nature and location of Ce cations.

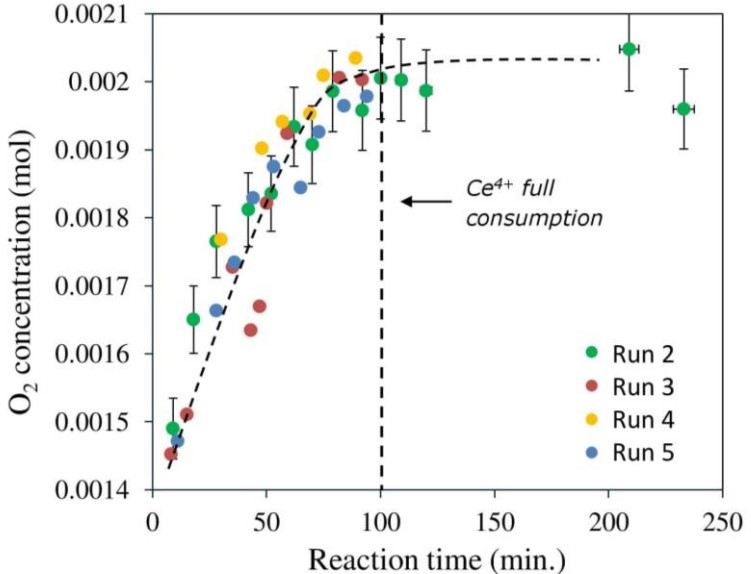

**Figure 11.** Repeated $O_2$ production as function of time using 4 wt.% $IrO_2/TiO_2$ and $Ce^{4+}$ cations as electron scavengers (0.182 Mol). Error bars are added on the first experiment only (but are similar for the remaining three runs). The lines are visual guides.

Figure S5 presents STEM, TEM and EDX images of the used catalyst (that was subject to five cycles of reaction). The STEM image (a) had two distinct areas, one of $TiO_2$ and one of $CeO_x$. It can also be seen that brighter spots were present only on $TiO_2$, which are attributed to $IrO_2$ clusters. The $IrO_2$ size distribution before and after the reaction revealed minor changes. Because $CeO_x$ was not found in close proximity to $IrO_2$ clusters, we conducted TEM and EDX analyses on a number of locations. These are presented in Figure S5b–d. In (b), a large area is presented that contains both $TiO_2$ and $CeO_x$, while (c) is only composed of $CeO_x$ with corresponding diffraction lines indicating the presence of $CeO_2$. Figure S5d presents EDX images of three regions (1, 2, and 3) from (b) and (c). It is clear that regions containing $CeO_x$ made no, or negligible, contributions of Ir, while only those composed largely of $TiO_2$ contained Ir. This may indicate that the interaction of $Ce^{4+}$ cations with $IrO_2$ clusters during the catalytic reaction does not result in direct precipitation at the local site, but rather that precipitation occurs over time, probably upon diffusion and agglomeration of $Ce^{4+}$ cations.

To further probe into the catalyst composition, XPS analysis of the used catalyst was conducted. The Ce3d and Ir4f spectra are given in Figure S6. Data for 1 wt.% $Ir/CeO_2$, which were prepared on purpose for further comparative studies and are worth inclusion, are also shown in the figure. Ce3d line was studied in detail, for both the $Ce^{4+}$ and $Ce^{3+}$ of $CeO_2$ and $CeO_x$, by many authors [39,40]. We and others previously showed that the deposition of noble metals such as Pd or Pt on $CeO_2$ often results in the formation of $Ce^{3+}$ cations because of the preferential removal of oxygen atoms at the $M/CeO_2$ interface [41–43]. We do not see the presence of $Ce^{3+}$ in the as-prepared catalyst (Figure

S6a); this is most likely because it is easier to oxidize Ir compared to Pd or Pt. Heating the catalyst in vacuum, however, resulted in a slight reduction, as evidenced by the appearance of the peaks at the V'/U' positions, which was attributed to the $Ce3d_{5/2,3/2}$ of $Ce^{3+}$. The XPS image for Ce3d of the spent $IrO_2/TiO_2$ is presented in Figure S6d. Because it involved ex situ measurements, it was not possible to obtain quantitative information related to the oxidation state of the Ce cations. It was, however, clear that the material was also composed of $Ce^{3+}$ cations (probably as $Ce_2O_3$) in addition to $CeO_2$. The ratio Ce to Ti was found to be about 0.2; this was an overestimation because Ce oxide particles were on the catalyst surface. There was, however, not much difference between Ir/Ti before and after reaction. This is in line with TEM data where $CeO_x$ were found to be away from (and not on top of) $IrO_2$ centers on $TiO_2$.

*2.5. Turnover Frequency Calculations*

Data obtained from TPR, XPS Ir4f and TEM were used to normalize the reaction rate in order to gauge which one is more suitable. The data are plotted in Figure 12. The $IrO_2$ particles in all catalysts studied were smaller than 2 nm in size. This means that the photoelectrons with a kinetic energy of about 1400 eV (XPS 4f) had the required escape depth to originate from the totality of atoms in the $IrO_2$ clusters dispersed on $TiO_2$. A similar argument applies to the TPR; despite their deviations, the signal originated from the totality of Ir clusters that were reduced to Ir metal. In addition, the same argument applies for the nominal amount of Ir used to prepare the catalysts. The figure shows that while there were deviations, these were mild and oscillated around similar TOF. This was not the case for TOF extracted from the surface atoms of $IrO_2$ clusters imaged by TEM. An example of how the surface atoms of Ir were extracted in $IrO_2$ clusters is given in Figure S7, in which they are assumed to be of hemispherical shape. In this case, the TOF seemed to increase with increasing loading. It is important to note that the $IrO_2$ clusters were of about the same size (between 0.7 and 1 nm or so) so any size effect would be small. The data then pointed to bulk atom participation in the redox reaction. This can be rationalized by the difference in diffusion rates of electrons and ions, with the latter being orders of magnitude slower than the former. It is worth indicating that while TOF values for Ir-based homogenous catalysts ranged from 30 to 45 min$^{-1}$ [15,17,18], and those in the case of heterogeneous $IrO_2$ catalysts were lower than the reported ones, these solid catalysts did not show signs of deactivation. The catalyst with 4 wt.% Ir loading was used for 5 consecutive runs at 0.182 M [$Ce^{4+}$] each time, which translated into a turnover number (TON) of ca. 56,000 (defined as total $O_2$ molecules (0.01 moles) over Ir atoms in the catalyst ($1.78 \times 10^{-7}$ moles)).

The slow kinetics of the reaction were likely behind its insensitivity to metal dispersion. Within the framework of Marcus Theory [44,45], a determining factor in the activation energy of the electron transfer between two reactants in solution is the so called "reorganization energy, $\lambda$", where the ions readjust their respective positions to optimize their orbital overlap for the transfer to occur [46]. This energy is computed as the difference in the redox potential of the two species involved in the reaction at the catalyst interface (strictly, the difference is $2\lambda$). This can be approximated as half of the energy difference between the oxidation potential of water (1.23 eV per electron) and that of $Ce^{4+}/Ce^{3+}$ [47], both with respect to NHE (as presented in Scheme 1, the redox potential of $Ir^{5+}/Ir^{4+}$ is within this region). Unlike thermally driven catalytic reactions, this relatively slow step makes the increase in dispersion unnecessary as long as electron transfer from the bulk to the surface of the catalyst occurs.

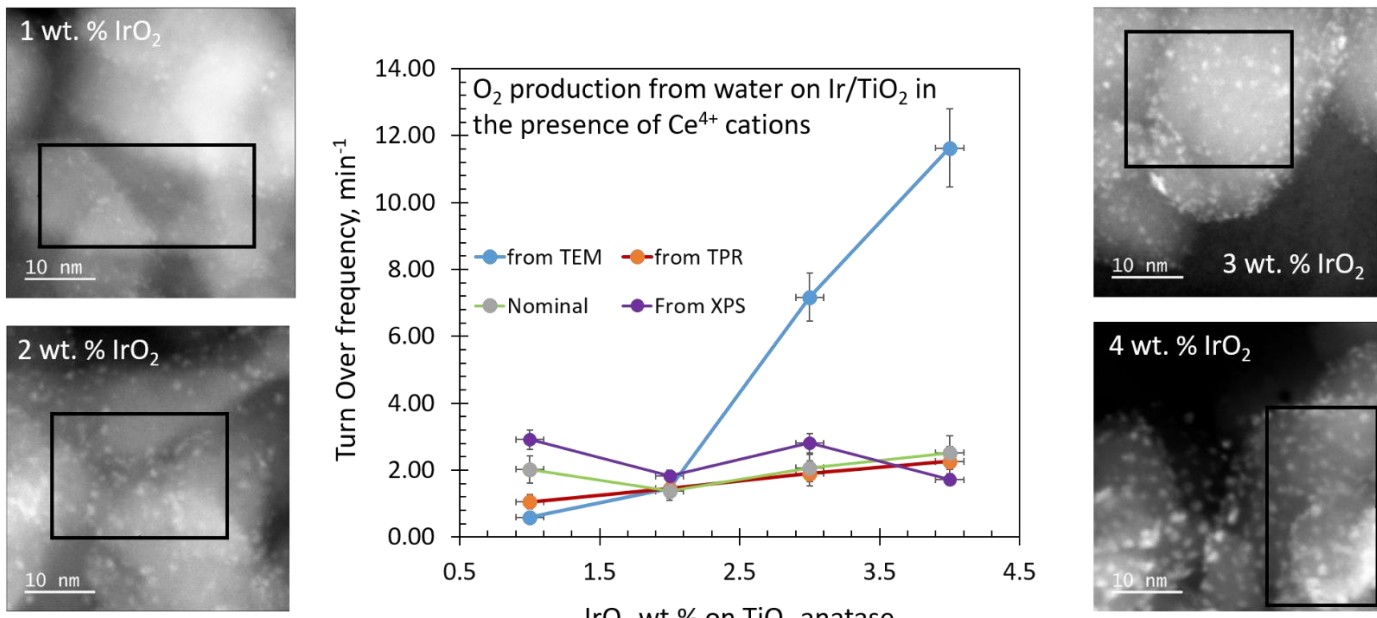

**Figure 12.** TOF using $IrO_2$-$TiO_2$ catalysts with 1, 2, 3, and 4 wt.%. TON was obtained by dividing the reaction rates at $t_{1/2}$ by the number of Ir atoms in the catalyst, as determined from the used amount of Ir in preparing the catalysts (nominal), from the total amount of hydrogen used to reduce $IrO_2$ to Ir (TPR), from the amount of Ir atoms determined from the elemental composition by XPS, and the number of surface atoms of Ir in $IrO_2$ clusters from TPR, assuming a spherical model, as described in the experimental section. The rectangles in the STEM images show the areas used to count the number of clusters in the four catalysts, from which their surfaces were extracted, assuming a hemispherical shape; deviations from these numbers were found to be negligible. The counting, within a few %, did not change from one area to the other of the same catalyst. The error bars are based on observed experimental deviations.

## 3. Experimental

### 3.1. Catalyst Preparation

Catalysts were prepared using the wet impregnation method. A quantity of 0.5 mg mL$^{-1}$ of $Ir^{3+}$ stock solution was prepared by dissolving pre-dried $IrCl_3$ x$H_2O$ (Sigma Aldrich, St. Louis, MO, USA) in deionized water (18.2 MΩ). The stock solution was then left under stirring overnight to ensure complete dissolution of the Iridium salt in the water. To prepare a catalyst, 2 g of anatase $TiO_2$ (Sigma Aldrich, St. Louis, MO, USA. BET surface area = 55 m$^2$/g) was heated at 100 °C for 10 min in a 500 mL beaker followed by the addition of a calculated volume of $Ir^{3+}$ from the stock solution, and sonication for 5 min to ensure effective mixing. The mixture was then heated at 110 °C on a hot plate under stirring at 160 rpm for 8 h until most of the liquid had evaporated. The hot plate top was fully covered with Al foil to ensure homogenous heating. The paste was then spread in a ceramic crucible and calcined at 400 °C (ramping rate: 2 °C min$^{-1}$) in a muffle furnace for 5 h.

### 3.2. Reaction Setup

All reactions were carried in a 140 mL flat bottom glass reactor sealed with a rubber septum under air assuming 21% $O_2$ concentration as a baseline. It was decided to use $O_2$ in air as a baseline as we found this method to be the most reproducible for measurements of molecular oxygen production. Briefly, a given amount of $(NH_4)_2Ce(NO_3)_6$ (CAN), ranging from 0.025 to 2 g, was dissolved in 20 mL DI water to form a yellow solution, followed by the addition of 20 mg of the catalyst. The final mixture, with a pH close to 1, was subjected to constant stirring throughout the reaction. A Gas Chromatograph (GC) (Thermo Fisher Scientific Inc., Waltham, MA, USA) equipped with a Thermal Conductivity Detector (TCD) was used for $O_2$ quantification using a 5 A column molecular sieve at 80 °C and He as a carrier gas. All catalytic reactions were run under the same conditions except for either the

use of catalysts with different Ir loadings or the use of $Ce^{4+}$ with different concentrations with a given catalyst. The pH decreased to below 1 after the reaction due to the release of protons; the exact value largely depended on the CAN concentration.

$Ce^{4+}$ cations in aqueous phase are present as $Ce(NO_3)_6{}^{2-}$; however, based on Raman spectroscopy data, the presence of dimers in the form of $(H_2O)_6Ce_2O(NO_3)_6$ was proposed [48]. The reaction rate for water oxidation would, therefore, additionally depend on the concentration of $Ce^{4+}$ cations. Water would need to dissociate on the surface, effectively giving pairs of hydroxyls, (**Ir-O** + HOH $\rightarrow$ HO-**Ir-O**-H) that are, therefore, abundant, and the rate could be considered zero order with respect to their concentration. This may not be the case for $[Ce^{4+}]$.

$$r = k\left[Ce^{4+}\right]^{n}$$

The reaction rate was proposed to be zero order with respect to $Ce^{4+}$ cations over Ir-based catalysts [15]. At $t = t_{1/2}$ and if $n$ is = 0, then $k = \frac{[Ce^{4+}]_0}{t_{1/2}}$, and if $n = 1$, then $k = \frac{0.693}{t_{1/2}}$. The $Ce^{4+}$ concentration during the progress of the reaction was monitored using $O_2$ generation according to Equation (1).

$$2H_2O + 4Ce^{4+} = 4Ce^{3+} + O_2 + 4H^+ \tag{1}$$

The turnover frequency (TOF) for each catalyst was calculated according to Equation (2).

$$\text{TOF}\left(\text{min}^{-1}\right) = \frac{Rate\ of\ O_2\ evolution\ at\ t_{\frac{1}{2}}\left(mol\ min^{-1}\right)}{(moles\ of\ Ir)} \tag{2}$$

Special attention was given to the determination of the moles of Ir that contributed to the reaction. Four different TOFs were computed: these were based on the nominal amount, and were derived from XPS Ir4f lines, from TPR and from HR-STEM. Details on the last three are given below.

### 3.3. Catalyst Characterization

#### 3.3.1. UV-Vis Absorbance

UV-Vis absorbance spectra of the powder catalysts were collected over a 250–900 nm wavelength range using a Thermo Fisher Scientific Evolution$^{TM}$ 300 UV-Vis spectrophotometer (Thermo Fisher Scientific Inc., Waltham, MA, USA) equipped with Praying Mantis$^{TM}$ diffuse reflection accessory purchased from Harrick Scientific (Harrick Scientific, Pleasantville, NY, USA). All spectra were recorded after $Ir/TiO_2$ filtration from the reaction mixture.

#### 3.3.2. X-ray Diffraction (XRD)

Powder XRD patterns of the samples were recorded on a Philips X'pert-MPD X-ray powder diffractometer (Malvern Panalytical, Malvern, UK). A 2θ interval between 10 and 90° was used, with a step size of 0.010° and a step time of 0.5 s. An Ni-filtered Cu $K_{\alpha}$ X-ray radiation source ($K_{\alpha}$ = 1.5418 Å) was operated at a current of 45 mA and a voltage of 40 kV.

#### 3.3.3. X-ray Photoelectron Spectroscopy (XPS)

XPS was conducted using a SPECS XRC 100 (SPECS Surface Nano Analysis GmbH, Berlin, Germany) equipped with a dual anode (Al and Mg) $K_{\alpha}$ X-ray source, and a double pass cylindrical mirror electrostatic energy analyzer (STAIB Instruments, Williamsburg, VA, USA). For each catalyst, a pellet of 10 mm diameter was prepared using 30–60 mg catalyst powder and an IR pelletizer at 10 tons of pressure for 45 min. The sample pellet was mounted on a molybdenum plate using spot welded tantalum wires, and loaded into an ultrahigh vacuum chamber with ca. $4 \times 10^{-10}$ torr base pressure. An electron beam was used to heat the sample when needed. The temperature measurements were performed using a pyrometer equipped with a laser pointer (Process Sensors Corp., Westborough,

MA, USA). The temperature measurements were taken from the sample plate of Mo with an emission coefficient of 0.27. The raw XPS data were processed using Casa XPS software (Version 2.3.16 PR 1.6, Casa Software Ltd., Teignmouth, UK, 2011). XPS Au4f at 84 eV was used for binding energy calibrations; Au foil was loaded prior to the $IrO_2$ pellets. For all measurements, Ir4f, O1s, C1s, and Ti2p were collected, and their peak areas were computed after correction to the photoionization cross section. Because of inevitable deviation from stoichiometry of the powder material, as well as the presence of non-negligible amounts of adventitious carbon, the ratio Ir to Ti was preferred for exact computation analysis. The used catalyst also contained Ce cations; in this case, Ce3d lines were also collected. To monitor the $Ce^{3+}$ cations, $Ir/CeO_2$ catalysts were independently prepared and heated inside the spectrometer chamber until some reduction of $Ce^{4+}$ cations to $Ce^{3+}$ cations occurred. In addition, to accurately monitor the Ir lines, we have measured XPS Ir4f of $IrO_2$ powder as a function of temperature inside the XPS spectrometer.

### 3.3.4. Temperature-Programmed Reduction (TPR)

TPR experiments were performed in a quartz tube coupled to a TCD (AutoChem 2920, Micrometrics, Norcross, GA, USA). All catalysts were purged with Ar for 1 h prior to the TPR experiments, which were performed under a constant flow of 10 vol.% $H_2$ in Ar mixture with a 10 °C min$^{-1}$ ramping rate and at a flow rate of 50 mL min$^{-1}$. The numbers of moles of reducible metal were calculated using the amount of $H_2$ consumed and a pre-determined $H_2$ calibration curve. $Ag_2O$ was used as standard for these. Several weights of $Ag_2O$ were used, from which a direct relationship between the signal and the amount of $H_2$ needed to reduce $Ag_2O$ to Ag was extracted. The amount of $H_2$ consumed was then computed for all $IrO_2$-based samples. The supporting information (Figure S1) contains more details.

### 3.3.5. Electron Microscopy (EM)

EM studies were performed using a Titan ST microscope (FEI company, Hillsboro, OR, USA), operated at an accelerating voltage of 300 kV, equipped with a field emission electron gun, a 4k × 4k CCD camera, and a Gatan imaging filter (GIF) Tridiem and Gatan imaging software suite (Gatan Inc., Pleasanton, CA, USA). The microscope was operated either in HR-TEM (phase contrast) or high angle annular dark field (HAADF)-STEM mode (Z-contrast) with a point-to-point resolution of ca. 0.12 nm and an information limit of ca. 0.10 nm in both cases. The HR-TEM beam focus was 100 nm while that of STEM was 1.0 nm. To prepare the samples, quantities of a few mg of catalyst were dispersed in ethanol followed by ultra-sonication of the mixture for 15 min. A drop of supernatant suspension was poured onto a holey carbon-coated Cu grid that was placed on filter paper and dried before the grid was loaded on to the sample holder. Catalyst nanoparticles that were sitting on holes were selected for analysis where possible. Further, Selected Area Electron Diffraction (SAED) and Energy Dispersive X-ray (EDX) analyses were performed on catalysts with exposed areas smaller than or equal to ca. 0.2 $\mu m^2$ to facilitate the microscopic analysis. SAED analysis was carried out in TEM while EDX was conducted in the STEM mode of operation.

### 3.3.6. Metal Dispersion

For spherical particles, a relationship between metal dispersion ($D$) and mean particle diameter ($d_m$) can be established by making assumptions regarding the nature of the crystal planes exposed on the metal surface. $IrO_2$ has a rutile structure and mainly exposes the (110) surface. The number of surface $IrO_2$ units per unit area ($n_s$) in the rutile $1 \times 1$ (110) plane lattice (3.16 Å × 6.38 Å) is $5 \times 10^{18}$ atoms m$^{-2}$ [49]. The surface area ($a_m$) occupied by an $IrO_2$ unit is simply the inverse of $n_s$. The volume ($V_m$) occupied by an Ir atom in the bulk was calculated using the $IrO_2$ atomic mass ($M$ = 224 g mol$^{-1}$), $IrO_2$

density ($p$ = 11.99 g cm$^{-3}$) and Avogadro's number ($N_A$ = 6.022 × 10$^{23}$ mol$^{-1}$), as given in Equation (3).

$$V_m = M/p \, N_A \qquad (3)$$

The specific surface area ($S_{sp}$) of a metal is related to its dispersion ($D$), as given in Equation (4).

$$S_{sp} = a_m(N_A/M) \, D \qquad (4)$$

$S_{sp}$ can also be given by Equation (5).

$$S_{sp} = (\Sigma n_i \, A_i/p \, (\Sigma n_i \, V_i)) \qquad (5)$$

$A_i$ is the surface area of an IrO$_2$ particle and $V_i$ is its volume. Based on TEM and STEM images of IrO$_2$ particles on the TiO$_2$ surfaces, we considered them to be spherical metal clusters. In this case, $A_i$ and $V_i$ became $1/2\pi d_i^2$ and $1/12\pi d_i^3$, respectively, where $d_i$ was the diameter of an IrO$_2$ particle. By substituting $A_i$ and $V_i$ in Equation (5), we obtained Equation (6).

$$S_{sp} = (6/p) \, (\Sigma n_i \, d_i^2/(\Sigma n_i \, d_i^3)) = 6/(p \, d_m) \qquad (6)$$

$\Sigma n_i \, d_i^3/\Sigma n_i \, d_i^2$ is also the mean particle size ($d_m$) of IrO$_2$ particles and was calculated using the particle size distributions revealed in STEM images. Equations (6) and (7), followed by the substitution of $V_m$ and $d_m$, allowed Equation (7) to be derived for the purpose of calculating metal dispersion.

$$D = 6 \, (V_m/(a_m \, d_m)) \qquad (7)$$

Equation (7) was used to calculate the Ir metal dispersion.

## 4. Conclusions

A wet impregnation method was used to prepare sub-nanometer IrO$_2$ clusters dispersed on top of TiO$_2$ anatase support (from 0.1 to 4 wt.%). The correlation between rates of reaction, Ir nominal wt.% loading and CAN concentrations indicated that IrO$_2$ clusters directly participated in the reaction as a catalyst. Changing the CAN concentration while keeping the same amount of Ir indicated a non-zero order dependence in the investigated range (Ce$^{4+}$/Ir$^{4+}$ between 876 and 7012). Although the TOF values in this case were lower than the reported values in homogenous systems, these catalysts did not show signs of deactivation. The catalyst with 4 wt.% loading was used for five consecutive runs, at 0.182 M [Ce$^{4+}$] each time, which translates to a turnover number (TON) of ca. 56,000 without changes in activity. Over a series of Ir/TiO$_2$ with Ir wt.% values equal to 1, 2, 3, and 4, the TOF calculations showed good similarities when extracted using the nominal wt.%, as well as those from TPR and XPS Ir4f results. The TOF numbers extracted using surface atom concentrations from HRSTEM were found to deviate; they increased with the amount of Ir. These results, in combination, indicate that all Ir atoms, in clusters of up to ca. 1 nm in size, participate in the electron transfer reaction for water oxidation.

**Supplementary Materials:** The following are available online at https://www.mdpi.com/article/10.3390/catal11091030/s1.

**Author Contributions:** Conceptualization, H.I.; methodology, M.A.; validation, M.A.N.; formal analysis, M.A., M.A.N., H.I.; investigation, M.A., K.A.W., M.A.N.; writing—original draft preparation, M.A., H.I.; writing—review and editing, M.A., H.I.; supervision, H.I.; project administration, H.I. All authors have read and agreed to the published version of the manuscript.

**Funding:** This research received no external funding.

**Data Availability Statement:** All data are available upon request from the authors.

**Conflicts of Interest:** All authors declare no conflicts of interest.

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
