# Peer review of "Extracting Turnover Frequencies of Electron Transfer in Heterogeneous Catalysis: A Study of IrO2-TiO2 Anatase for Water Oxidation Using Ce4+ Cations"

_catalysts, doi:10.3390/catal11091030_

Round 1

Reviewer 1 Report

The authors prepared a heterogeneous catalyst system composed of IrO2 with a mean particle size ranging from 0.5 to 1 nm dispersed on a TiO2 anatase support for water oxidation using Ce4+ cations. The manuscript is well organized and therefore I recommend the publication of this paper with minor revision, while the following issues need to be improved.

  1. EDS mapping images for the IrO2/TiO2 catalyst with 4 wt% should be given and discussed.
  2. Whether the Ce4+ cations can be inserted into the catalyst of IrO2/TiO2 during the water oxidation reaction.
  3. The authors should check the full manuscript for the mistakes (Line 452, Page 14; Line 620, Page 20; etc.)

Author Response

Referee 1

The authors prepared a heterogeneous catalyst system composed of IrO2 with a mean particle size ranging from 0.5 to 1 nm dispersed on a TiO2 anatase support for water oxidation using Ce4+ cations. The manuscript is well organized and therefore I recommend the publication of this paper with minor revision, while the following issues need to be improved.

Thank you

1. EDS mapping images for the IrO2/TiO2 catalyst with 4 wt% should be given and discussed.

The EDS of the fresh 4 wt.% Ir catalyst are given in figure 1. Those of the used one are in figure S5.

2. Whether the Ce4+ cations can be inserted into the catalyst of IrO2/TiO2 during the water oxidation reaction.

Yes, Ce4+ can be inserted into the catalyst during the reaction. The five consecutive runs were conducted that way (figure 11)

3. The authors should check the full manuscript for the mistakes (Line 452, Page 14; Line 620, Page 20; etc.)

Thank you, corrected, the manuscript was further checked for mistakes.

Reviewer 2 Report

The present manuscript investigates the effect of IrO2 loading on TiO2 and their particle size distribution for oxygen evolution reaction using the CAN redox system. The study emphasizes a heterogenized Ir-based system followed by TON calculation to compare with homogeneous oxygen evolution catalysts. IrO2/TiO2 was prepared by the wet impregnation method with different IrO2 loading. TEM, STEM, XRD, XPS and TPR well-characterized the resulting catalyst. OER using 4 wt.% IrO2 on TiO2 demonstrates almost identical performance even after 5 runs with a TON of 56000. The obtained results are explained engagedly. It is heartening to read the TPR explanation. The manuscript can be accepted for publication after a few minor revisions.

  1. A nicely sketched graphical abstract should be added.
  2. The introduction is a bit long and can be reduced by removing some non-relevant content.
  3. The motivation of the study is not clear at the initial stage and needs significant attention. To attract the readers' attention, it is recommended to add motivation and key findings at the end of the introduction part.
  4. The journal's name in the references must be abbreviated.
  5. XPS peaks deconvolution is not correct and must be corrected. A dedicated XPS program must be used for peak deconvolution.
  6. Figure numbering is not correct. STEM image in (Figure 1. (A) ) correct Figure 5A must be clear.

Author Response

Referee 2

The present manuscript investigates the effect of IrO2 loading on TiO2 and their particle size distribution for oxygen evolution reaction using the CAN redox system. The study emphasizes a heterogenized Ir-based system followed by TON calculation to compare with homogeneous oxygen evolution catalysts. IrO2/TiO2 was prepared by the wet impregnation method with different IrO2 loading. TEM, STEM, XRD, XPS and TPR well-characterized the resulting catalyst. OER using 4 wt.% IrO2 on TiO2 demonstrates almost identical performance even after 5 runs with a TON of 56000. The obtained results are explained engagedly. It is heartening to read the TPR explanation. The manuscript can be accepted for publication after a few minor revisions.

Thank you, it was a trying time to describe the TPR results.

1. A nicely sketched graphical abstract should be added.

Added

2. The introduction is a bit long and can be reduced by removing some non-relevant content.

The introduction starts with the use of Ce4+ cations to capture electrons, then introduces the catalytic systems. These are followed by an introduction of TOF numbers, how they are calculated and used in heterogeneous catalysis and the fact that it is not a straight forward exercise. We would like to keep these in the introduction.

3. The motivation of the study is not clear at the initial stage and needs significant attention. To attract the readers' attention, it is recommended to add motivation and key findings at the end of the introduction part.

We have added in the last paragraph of the introduction more on the motivation.

4. The journal's name in the references must be abbreviated.

OK, abbreviated, thank you

5. XPS peaks deconvolution is not correct and must be corrected. A dedicated XPS program must be used for peak deconvolution.

XPS data are analyzed using Casa-XPS (mentioned in the experimental section) with additional information given in figure S2.

6. Figure numbering is not correct. STEM image in (Figure 1. (A) ) correct Figure 5A must be clear.

The numbers of the figures have been corrected.